# An Evaluation of a Low-Intensity Cognitive Behavioral Therapy mHealth-Supported Intervention to Reduce Loneliness in Older People

**DOI:** 10.3390/ijerph16071305

**Published:** 2019-04-11

**Authors:** Mary Ann Jarvis, Anita Padmanabhanunni, Jennifer Chipps

**Affiliations:** 1School of Nursing and Public Health, Desmond Clarence Bldg. Howard College Campus, University of KwaZulu-Natal, Durban 4001, South Africa; jarvism@ukzn.ac.za; 2Department of Psychology, University of the Western Cape, Robert Sobukwe Avenue, Belville 7535, South Africa; apadmana@uwc.ac.za; 3School of Nursing, Faculty of Community Health, University of the Western Cape, 14 Blanckenberg Road, Belville 7535, South Africa

**Keywords:** older people, loneliness, mHealth, low-intensity CBT, smartphone, WhatsApp

## Abstract

There is a high prevalence of loneliness among older people, especially in residential care settings. Loneliness is often accompanied by maladaptive cognitions which can affect the maintenance and establishment of meaningful social connections. This study implemented and evaluated a low-intensity Cognitive Behavior Therapy (LI-CBT) mHealth-supported intervention which targeted maladaptive cognitions in older people (≥60 years) experiencing loneliness. The three-month intervention using WhatsApp was implemented with older people in four inner-city residential care facilities. The intervention included three components: technology acceptance, psycho-education, and individualized positively worded messages addressing maladaptive cognitions. The intervention was evaluated using a randomized control design. Key outcomes were measured pre-, post-, and one month after the intervention. There were significant changes in social cognition (YSQ-SF T_0_–T_1_–T_2_, *X*^2^ = 9.69, *p =* 0.008) and loneliness levels (total loneliness T_0_–T_1_–T_2_, *X*^2^ = 14.62, *p* = 0.001), and an increase in WhatsApp usage (T_0_ = 26% vs. T_1_ = 60%, *X*^2^=15.22, *p* = 0.019). At 1-month follow-up, even with a significant reduction in WhatsApp usage, a significant reduction in loneliness was maintained. LI-CBT delivered via instant messaging may be effective in reducing loneliness experienced by older people.

## 1. Introduction and Background

Relocation to residential care may bring about changes in social network structures and social network density for older people [1,2,3]. This can result in social isolation, loneliness and reduced mental well-being [4,5], with loneliness prevalence reported as high as 56% among older people living in residential care in Norway [6]. Loneliness, seen as the cognitive discrepancy between desired and actual quality and quantity of relationships [7], may result in negative or maladaptive cognitions commonly expressed by people experiencing loneliness [8,9] that can interfere with the development or maintenance of social connections [8,10]. Loneliness has pathophysiological effects on older people which increase mortality [11].

There are a plethora of primary studies, and systematic reviews addressing loneliness in older people [5,12,13]. Various social and psychological interventions have been recommended to address loneliness in older people, including interventions using technology [5,9,14,15], with varying degrees of effectiveness [16] and differing levels of quality in evaluation design [8,9]. Interventions addressing maladaptive or negative social cognitions have been reported to be most effective in reducing loneliness by changing social behavior and thereby improving social connections [8,9]. Improvement in social connections can contribute to healthy ageing and improved cognitive functioning [17,18]. Previously held beliefs that face-to-face interaction is the gold standard of social participation [19] has been challenged, with evidence that feelings of connection can also be established via communication technology [3]. Social media and instant messaging have been cited as instances of communication technology that may improve social networks by maintaining visibility, re-awakening previous social networks, or development of new networks [20,21,22,23]. Instant messaging using mobile cellular phones can be used to address maladaptive cognitions, but older people are reported to be less inclined to use technological communication to initiate social connections [24]. MHealth projects for older people are often viewed with skepticism in regard to cost, perceived poor technology acceptance and ageism [25,26], but when appropriate consideration is given to facilitating conditions [25,27,28] and cost [29,30] older people are capable of learning new technology [25]. Despite studies examining older people’ existing engagement with social media and its effects on loneliness [21,31] a gap exists for intervention studies that specifically target loneliness using social media. Chipps et al. [14] in their umbrella review of systematic reviews examining eInterventions targeting loneliness in older people failed to identify any mHealth interventions in the included primary studies. Chen and Schulz [12] in their systematic review of communication technology interventions, suggested further studies in the fields of loneliness and social media to identify the opportunities to reduce loneliness in older people. This study accordingly sought to implement and evaluate a low-intensity cognitive behavioral therapy (LI-CBT) mHealth-supported intervention delivered via mobile instant messaging (WhatsApp) to address maladaptive cognitions and reduce loneliness in older people living in a residential setting.

## 2. mLINCC—A LI-CBT Intervention Supported by WhatsApp

A LI-CBT intervention, *Living In Network Connected Communities* (mLINCC) to be delivered on WhatsApp, was developed by a stakeholder group consisting of a CBT specialist psychologist, a mental health nurse, and an mHealth expert. The intervention was informed by a theoretical framework of loneliness [32,33] and evidence from two umbrella reviews on the effectiveness of interventions to address loneliness [14,34]. A gap in mHealth interventions was identified [14,23] along with strong evidence that social cognition interventions [8,9] and psycho-education [35] were most effective in addressing loneliness and the need for participants to be actively involved in the intervention [35].

The three-month intervention had three distinct phases involving threads of trust, social network development, and gerontechnology self-efficacy, reinforced with homework assignments requiring a WhatsApp response [36] and a weekly face-to-face Help Desk for technological support [37]. All participants in the intervention group (IG) were added to the mLINCC WhatsApp group, which was a closed group and at cross-over a separate WhatsApp group (mLINCC 2) for the control group (CG). Apart from the closed groups, participants were able to use the smartphone and communication on mLINCC as they chose and there was no incentive for increased use. The data package selected to offer the greatest financial sustainability post-intervention, allowed for any form of WhatsApp interaction apart from WhatsApp calls. No participants had previously used WhatsApp; in the IG apart from one participant all owned feature (basic) mobile phones which did not support such applications as WhatsApp, predominantly making calls and the participant who owned a smartphone used it as a feature (basic) mobile phone. In the first phase (2 weeks), technology acceptance was developed through activities which included the selection of a smartphone and four 90-min face-to-face (F2F) group-training sessions on the use of the smartphone and WhatsApp. The participants could keep the phones post-intervention but would be responsible for the purchase of data (600 Mb for ZAR15.00/month/US $1.10). In the second phase (2 weeks), psycho-education was delivered through four 90-min F2F sessions on factors underlying loneliness. Key messages from these sessions were confirmed through messages on WhatsApp. In phase three (1 month) individualized messages with positively framed content aimed at countering maladaptive cognitions were sent to participants via WhatsApp. Individual maladaptive cognitions were identified through the administration of the Young Schema Questionnaire short-form (YSQ-SF)—Disconnection and Rejection domain. The messages were constructed by the CBT specialist and each message had two components: acknowledgment of the maladaptive cognition linked to loneliness, and a positively framed counter message to allow the participants an opportunity to reflect on the cognitive distortion and its influence on their feelings and behaviors (Table 1). The final phase was a maintenance stage (1 month) during which just the weekly Help Desk was continued, with the moderator still presiding over the WhatsApp group though not participating.

## 3. Methodology

*Setting:* The study was conducted in an Non-Government Organization (NGO) which offers accommodation for resource-restricted residents, ranging from independent living to frail care and services throughout the city to 1900 older people (≥60 years).

*Study design:* A randomized control study with two groups—IG and CG—was conducted to evaluate the intervention. Due to the nature of the intervention there was no blinding. Participants were randomly assigned to the IG or the CG. The CG received usual care, which was a generic wellness program for all residents. For ethical reasons, on completion of the intervention the CG also received the intervention, receiving their smartphones and all resources at this cross-over point. The study was approved by the Humanities and Social Sciences Ethics Committee (HSSREC) of the University of KwaZulu-Natal. (HSS/1169/016D) on 16 September 2016, followed by gatekeeper permission from the research site. Approval was received to feature the WhatsApp brand in the printed material (Asset Request # 34292).

*Population and sampling:* The population was 828 residents living in four inner-city residential NGO care facilities (*n* = 828) in a large South African city. Eligible participants were identified through a cross-sectional survey of a random sample of 277 residents and met the criteria for inclusion: namely, ≥60 years, residing in the residence, willing to participate, cognitively intact on the Subjective Memory Complaint Clinical (SMCC) [38] and socially isolated (≤15 on the Friendship Scale) [39], or lonely (2–6 on de Jong Gierveld Loneliness Scale (DJGLS) [40] [6-item]) or with decreased mental well-being (World Health Organization-Five Well-Being Index, (WHO-5) [41] score < 13) (Figure 1) A minimum sample size of 52 participants was estimated to provide a medium to large effect size. 

*Research instrument*: A standard researcher-administered questionnaire was used to capture the key outcome measurements of WhatsApp usage, social cognitions, loneliness, and mental well-being. The questionnaire captured data on demographics and WhatsApp usage, social cognition (Disconnection and Rejection domain of the Young Schema Questionnaire (YSQ-SF 3)) [42], loneliness (6-item) (DJGLS) [40], and mental well-being (WHO-5) [41] (Table 2).

*Data collection*: The questionnaire was administered three times: pre-intervention (T_0_), post-intervention (T_1_) and one month post-intervention (T_2_). The participants provided written informed consent and the LI-CBT mHealth-supported intervention was conducted from March to June 2018. In preparation, research assistants (1:4 participants) were trained and resource materials prepared (manuals, stylus pens, and phone pouches) [48,49,50].

*Data analysis:* Data from the questionnaires were entered into SPSS v.25. Ages were categorized into younger old (60–79 years) and older old (80+ years) and time in residence (1–18 months and ≥19 months). Summary scores for all outcome measures were calculated. Items in DJGLS were reverse scored for emotional loneliness subscale. Non-parametric Friedman test (*X*[2]) was used to detect within-group differences in the intervention across three time periods tests, followed by Wilcoxon signed rank test (*W*) to compare the time points for significance (*p ≤* 0.05). Differences between groups were compared using Mann-Whitney-U (*U*) and due to the small sample and different sample sizes, Hedges’ g calculated effect sizes ([*M*_1_–*M*_2_]⁄*SD*_pooled and weighted_) (https://www.socscistatistics.com/effectsize/default3.aspx). Missing data was handled by excluding cases pairwise. Cronbach’s alpha was used to test the internal consistency of the subscale and scale scores (≥0.70 good).

## 4. Results

### Demographics

Eighty-nine respondents (*n* = 89) were eligible to participate in the intervention, and 32 (35.95%) agreed to participate (Figure 1). Of the eligible participants 32 (35.95%) were not interested, 13 (14.61%) ill, 10 (11.24%) were not available and two participants declined for personal problems (Figure 1), Eligible participants who declined participation showed similar levels of mental well-being (WHO-5 *m* = 16.5/25 ± 4.9) and total loneliness (*m* = 3.14/6 ± 1.59) to the participants. Fifteen participants were allocated to the IG and 17 to the CG, resulting in 13 (IG) and 16 (CG) participants after attrition (Figure 1). The participants were between 61 and 87 years (*m* = 74.93 ± 6.41 years), mainly female, Asian/Indian, widowed, and primary school and lesser-educated. Nearly two thirds of the participants had lived in the residences for 19 months or longer (*n* = 20, 62.5%) (Table 2). Cronbach-alpha was measured at T_0_ and shown to be good for YSQ-SF and all five schemas and the WHO-5, but low internal constancy for social loneliness (α = 0.55).

At T_0_ there were no significant demographic or outcome differences between the IG and CG (Table 3). At the start of the intervention, only slightly more than half of participants (54.84%) used their mobile phones weekly to contact family or friends (Table 4), with the majority (*n* = 26, 83.87%) using the call function. The total group (*n* = 32) were mostly mentally well, but showed moderate levels of maladaptive social cognitions on the selected domain (YSQ-SF *m* = 78.38/150 ± 25.06), with *Social Isolation* rated the highest, and moderate levels of loneliness (total loneliness *m* = 3.56/6 ± 1.24; emotional loneliness *m* = 1.63/3 ± 0.83; social loneliness *m* = 1.94/3 ± 0.72) (Table 4).

The effectiveness of the intervention was evaluated at three time points (T_0_–T_1_–T_2_). In the IG at T_0_–T_1_–T_2_, a significant increase in the frequency of mobile phone usage (*X*^2^ = 8.90, *p* = 0.012) and WhatsApp (*X*^2^ = 12.17, *p* = 0.002) was recorded, as opposed to the CG where no significant changes in the frequency of usage of the mobile phone were recorded. In the IG from T_1_ to T_2_, the significant increase in the frequency of daily smartphone usage (T_0_ = 26% vs. T_1_ = 60%, *X*^2^ = 15.22, *p* = 0.019) was not maintained with a significant reduction by the end of the intervention (T_1_ = 60% vs. T_2_ = 53.3%, *X*^2^ = 13.48, *p* = 0.036). Monitoring of participants’ smartphone activity was only possible through the WhatsApp group (mLINCC) and not through their encrypted messages. On mLINCC there was evidence of participants’ active use of all functions offered by the select WhatsApp data package (excluded WhatsApp calls), except for the function of creating a voice message. While recording a voice message and the simultaneous requirement of holding the voice message icon appeared to present co-ordination challenges.

Over the intervention (T_0_–T_1_–T_2_), the IG showed significant decreases in the total YSQ-SF (*X*^2^ = 9.69, *p =* 0.008) and the schemas (*Social Isolation X*^2^ = 16.45, *p ≤* 0.001, *Emotional Deprivation X*^2^
*=* 11.04, *p* = 0.003, *Defectiveness X*^2^ = 9.50, *p* = 0.009) except in the schemas *Abandonment* and *Mistrust* (Table 5). This was accompanied by a significant reduction (T_0_–T_1_–T_2_) in total loneliness (*X*^2^ = 14.62, *p* = 0.001) and emotional loneliness (*X*^2^ = 15.80, *p* = < 0.001). In comparison. (T_0_–T_1_–T_2_), the CG participants became more lonely (total loneliness, *W* = 5.51, *p* = 0.064) and significantly more emotionally lonely (*X*^2^ = 12.29, *p* = 0.002) (Table 5). For the IG, T_0_ to T_1_ showed the most significant reductions for all measurements (YSQ-SF (YSQ-SF *W* = −2.34, *p* = 0.019, Social *Isolation W* = −3.06, *p* = 0.002, *Defectiveness W* = −2.72, *p* = 0.007, *Emotional Deprivation W* = −2.45, *p* = 0.014, Abandonment *W* = −2.08, *p* = 0.037, total loneliness W = −2.16, *p* = 0.031, emotional loneliness *W* = −2.27, *p* = 0.023)) (Table 5). During the post-intervention period (T_1_–T_2_), the CG had significantly higher loneliness (total) (*p* = 0.011) and emotional loneliness (*p* = 0.003) (Table 5).

Comparing the overall effect between the IG and CG pre- and post-intervention (T_0_–T_1_) the IG had significantly improved WhatsApp usage (IG 73.3% vs. CG 11.8%, *X*^2^ = 16.22, *p* ≤ 0.001), with accompanying significantly greater improvements in their social cognitions for *Social Isolation* (IG 7.08 ± 5.62 vs. CG 1.29 ± 6.08, *U* = −2.39, *p* = 0.015), *Emotional Deprivation* (IG 6.69 ± 7.58 vs. CG 1.12 ± 7.75, *U* = −2.24, *p* = 0.025), *Abandonment* (IG 6.69 ± 7.58 vs. CG 1.12 ± 7.75, *U* = −2.24, *p* = 0.025). The improvements in loneliness and mental well-being were not significant between the groups at T_1_ (Table 6). However, in the post-intervention period (T_1_–T_2_), the significant increase in WhatsApp users (IG 66.7% vs. CG 5.9%, *X*^2^ = 17.08, *p ≤* 0.001), was accompanied by significant improvements in the IG compared to the CG in all levels of loneliness (Total loneliness IG 0.92 ± 1.26 vs. CG-1.53 ± 2.00, *U* = −3.28, *p* = 0.001; emotional loneliness IG 0.31 ± 0.75 vs. CG 1.12 ± 1.11, *U* = −3.36, *p* = 0.001; social loneliness (IG 0.62 ± 1.04 vs CG −0.41 ± 1.18, *U* = 2.36, *p* = 0.028) (Table 5). The largest effect size was for total loneliness and emotional loneliness (total loneliness *ES* = 1.50, emotional loneliness *ES* = 2.26) (*p* ≤ 0.001) (Table 6), with small effect sizes for social loneliness (*ES* = 0.47) and mental well-being at T_2_ (*ES* = 0.41) (Table 6).

## 5. Discussion

Although the study sample size was small, the LI-CBT WhatsApp intervention was effective in significantly adjusting maladaptive social cognitions and reducing loneliness in older people in this setting and can be used in other resource-restricted settings and non-resource-restricted settings. Customization of the intervention is suggested in tailoring the positively worded MIM to be culturally and contextually specific. In future at the invitation to the intervention a period of experimentation with a smartphone in a low risk setting has the possibility of increasing enrolment. The study though delivered using WhatsApp concurred with the study byChiang et al. [15] involving reminiscence in older men (*n* = 92) which showed the effectiveness of targeting maladaptive social cognitions in the reduction of loneliness.

Improved cognition and a reduction in loneliness was significantly evident post-intervention (except social loneliness) and was maintained one month after the active intervention. The continued effect, despite some decrease in WhatsApp usage possibly due to the lessening of the novelty effect [51], may be related to the changes in the format and frequency of communication with the social networks from participants’ initial preference for mobile phone calls to their use instead of WhatsApp, which allowed them to maintain or develop contact with both kin and non-kin networks.

The social cognition schema which demonstrated the largest effect size (ES = 1.28) was *Social Isolation*, with changes in the feelings of alienation, not belonging, being an outsider, and socially different. *Social Isolation* has predictive validity for depression [52], and links to emotional loneliness [53]. Similarly, deprivation is a core component of loneliness [7] and the large effect on *Emotional Deprivation* (ES = 1.24), which involved improved feelings of nurturance, empathy and protection may further reduce emotional loneliness [53]. Improvement in the schemas of *Emotional Deprivation*, *Social Isolation*, and emotional loneliness were paralleled with an increase in smartphone contact with social networks. The findings suggest that through the network link in the intervention, the participants found nurturance and a sense of belonging and inclusion, and were able to experience relationship salience [19]. 

The effect for social loneliness (ES = 0.47) was small and appeared more resistant to change. The IG participants showed slower changes in their perceptions of adequacy in the number of people in the social networks and it was only after the maintenance period that social loneliness showed a near significant change. This may relate to the progressive nature of loneliness, with dissipation of emotional loneliness preceding dissipation of social loneliness [54]. It was anticipated that social loneliness improvement would have occurred earlier due to the early face-to-face interaction among participants and the group-based interventions, reported to be of value in the reduction of loneliness [9,32]. The resistance to change in social loneliness may also relate to the resistance to change in the schema of *Mistrust*, needing a change to occur in the cognitive appraisal of relationships before there is a change in the social deficit [54]. The cognitive appraisal of *Mistrust* can be a problem for social engagement: trust, which is often likened to the “glue” that binds society [55], is closely intertwined with social loneliness. Edwards [56], following Putnam’s various propositions in relation to social capital, states that trust is foundational to civic culture and is the outcome of a person’s correct prediction of others’ co-operative behavior, facilitated by external conditions. In addition, Pope et al. [57] note that the higher the level of trust, the greater the possibility of social participation. Because *Mistrust* is linked to social loneliness it may also be interwoven with the schema of *Defectiveness* leading to feelings of shame, inadequacy, and hypersensitivity to criticism that hamper trust and relationship development. A further possible factor relates to socio-emotional selectivity theory, according to which older people select to maintain contact with their closest, familiar relationships [57] rather than building new relationships [58,59]. Lastly, delayed changes in social loneliness, may be related to South Africa being a country with low levels of trust [54] and with older people showing lower trust levels than younger people [55]. Mistrust is often higher in settings such as South Africa that have high levels of crime, social inequalities, low income and political corruption [55].

## 6. Conclusions

The unique LI-CBT mHealth supported by the social networking platform of WhatsApp (mLINCC) incorporated face-to-face development of technological readiness in smartphone usage, psycho-education, and the delivery of positively worded mobile instant messages. Overall the intervention showed significant improvements in maladaptive cognitions, and loneliness, highlighting the role of cognitive appraisals in the subjective experience of loneliness. In addition, the unique use of WhatsApp and the crafting of a LI-CBT mHealth-supported intervention that addressed technological readiness in smartphone usage mitigated against loneliness experienced by cognitively intact older people living in residential care settings. The study demonstrated the possibility of the delivery of LI-CBT by a non-psychologist in a low resourced setting in Africa and demonstrated the ability of older people to learn smartphone technology thereby facilitating e-inclusion.

## 7. Recommendations

It is recommended that residential care facilities implement measures to facilitate residents’ positive cognitive appraisals of relationships and contexts such as regular psycho-education sessions. It is further recommended that the study is repeated with a larger sample, over for a longer period, possibly six months, and involves community dwelling older people. In particular, this should be conducted in Africa, where there is a paucity of comparable loneliness studies. In addition, further studies on the development of trust among older people could result in a measure to increase enrolment in intervention programs targeting loneliness.

## 8. Limitations

Overall, the study found that a low-intensity CBT intervention, tailored to individuals, can be delivered to older people using smart phones and instant messaging with potential to reduce loneliness, although some limitations in the study should be noted. Firstly, the sample size was low, with eligible participants showing low intent to participate (64.04%), which may be a side-effect of being lonely and may require different study recruitment in future studies. This is confirmed by high attrition rates in loneliness studies [51,60,61,62] due to higher levels of physical ill-health in older people experiencing loneliness [63], coupled with reticence to use smartphone technology possibly due to anxiety levels or a reduced sense of self-efficacy [27]. The low number of participants in the study and the high number of demographics variables and outcome measures could have had an impact on significance. Secondly, a limitation in the use of the DJGLS in this setting should be noted, with possible misinterpretation by the participants of the terms “plenty”,” many” and “enough” in the DJGLS [64] that may have influenced ratings of social isolation, and also the notably low reliability in the social loneliness subscale for this study in comparison with other studies [40]. Thirdly, the nature of the study design did not enable differentiation between the various components or the vehicle of delivery of the intervention, and it is recommended that factorial and smart designs be used for future evaluations.

## Figures and Tables

**Figure 1 ijerph-16-01305-f001:**
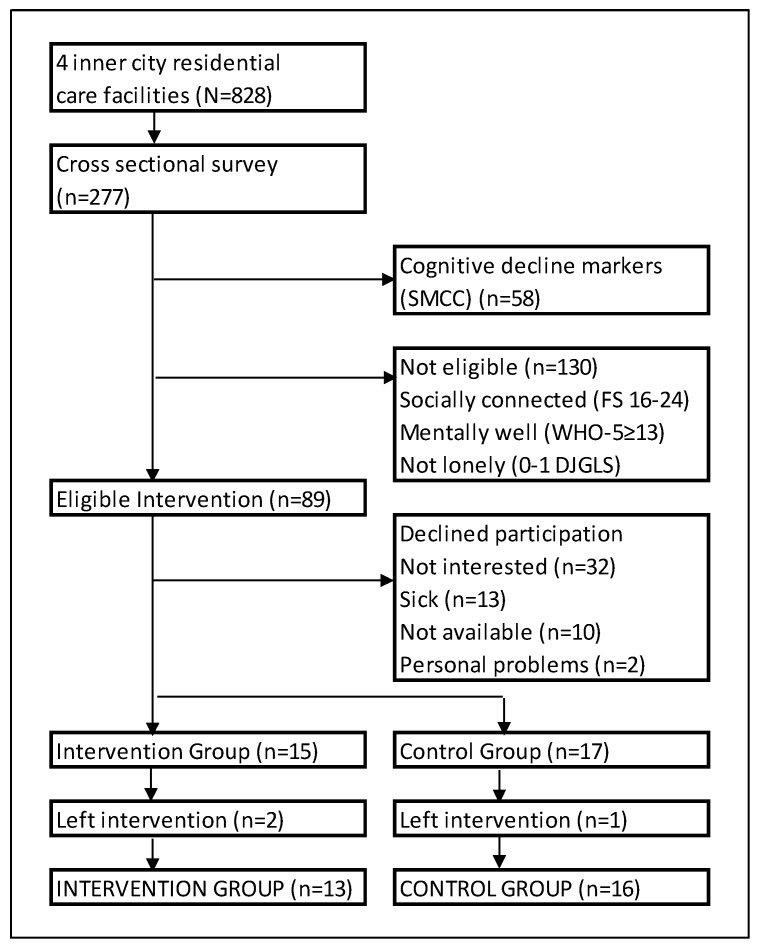
Participant allocation and attrition. DJGLS: de Jong Gierveld Loneliness Scale; FS = Friendship Scale; SMCC: Subjective Memory Complaint Clinical; WHO-5: WHO (five) Well-being Index.

**Table 1 ijerph-16-01305-t001:** Example of Mistrust schema and a positively framed message.

Item on YSQ-SF	Acknowledgment of Schema (Day 1)	Positively Framed Message Delivered on mLINCC (Voice Note and Text) (Day 2)
I am quite suspicious of other people’s motives	It is understandable that you may feel suspicious of other people and worry that they may hurt you, take advantage of you, or have ulterior motives, especially if these types of experiences have happened in the past.	Counter message; Not all people have negative intentions. There are genuine and trustworthy people in the world. Unfortunately, negative expectations can act as a barrier to your being able to reach out to others. You can change this. Try to evaluate the accuracy of your beliefs. Look back on your life and identify someone who really cared about you. Think about what made this person different.
It is only a matter of time before someone betrays me		
I feel that people will take advantage of me		
I feel that I cannot let my guard down in the presence of other people, or else they will intentionally hurt me		
I am usually on the lookout for other people’s ulterior or hidden motives		

**Table 2 ijerph-16-01305-t002:** Outcome measures.

Measure	Description
Usage	Frequency of mobile phone use to contact family or friends (less than monthly, monthly, weekly, or daily). Predominant function used on mobile phone.
Social Cognition	Disconnection and Rejection domain and related schemas of the YSQ-SF (Abandonment, Defectiveness, Emotional Deprivation, Mistrust and Social Isolation) [43]. Using a 6-point Likert scale rating the most accurate description over the past year from “Completely untrue of me” (1) to “Describes me perfectly” (6) [43,44,45]. YSQ-SF has established reliability and validity in Eastern (Cronbach α 0.72 –0.90) and Western settings (Cronbach α 0.81–0.94) [44] and with the elderly [45].
Loneliness	The DJGLS is a 6-item scale (0–6) with two sub-scales of emotional (0–3) and social loneliness (0–3) [40]. A score of six represents extreme loneliness and three of extreme emotional or social loneliness, measured on its sub-scales [40]. It is recommended for use in the elderly [46], reliable and well validated [40].
Mental well-being	The WHO-5 is a 5-item scale using a 6-point Likert rating and is a first screener for depression (raw score <13) [41,47]. The scale items are positive statements based on mood, vitality and general interest over a two-week period [41].

**Table 3 ijerph-16-01305-t003:** Baseline demographics (*n* = 32).

Variable of Interest	Total Participants (*n* = 32)	IG *n* = 15 (46.88%)	CG *n* = 17 (53.13%)	Statistic	*p*-Value
Age group				*X*^2^ = 0.41	*p* = 0.522
Younger old (60–79)	27 (84.4%)	12 (80.0%)	15 (88.2%)		
Older old (80+)	5 (15.6%)	3 (20.0%)	2 (11.8%)		
Gender				*X*^2^ = 0.54	*p* = 0.659
Males	6 (18.8%)	2 (13.3%)	4 (23.5%)		
Females	26 (81.3%)	13 (86.7%)	13 (76.5%)		
Marital status				*X*^2^ = 1.31	*p* = 0.726
Never married	7 (21.9%)	3 (20.0%)	4 (23.5%)		
Married	5 (15.6%)	3 (20.0%)	2 (11.8%)		
Separated/divorced	9 (28.1%)	3 (20.0%)	6 (35.3%)		
Widowed	11 (34.4%)	6 (40.0%)	5 (29.4%)		
Highest Education				*X*^2^ = 1.00	*p* = 0.755
Pre-primary and lower	18 (56.3%)	8 (53.3%)	10 (58.8%)		
2° and 3° education	14 (43.8%)	7 (46.7%)	7 (41.2%)		
Time living in res				*X*^2^ = 0.21	*p* = 0.647
1–18 months	12 (37.5%)	5 (33.3%)	7 (41.2%)		
≥19 months	20 (62.5%)	10 (66.7%)	10 (58.8%)		
Children				*X*^2^ = 0.03	*p* = 0.589
No children	9 (28.1%)	4 (26.7%)	5 (29.4%)		
Yes children	23 (71.9%)	11 (73.3%)	12 (70.6%)		
Grandchildren				*X*^2^ = 0.74	*p* = 0.388
No grandchildren	11 (34.4%)	4 (26.7%)	7 (41.2%)		
Yes grandchildren	21 (65.6%)	11 (73.3%)	10 (58.8%)		

CG: Control group; dep: IG: Intervention group; 2°: Secondary; 3°: Tertiary; Note: Differences in demographic variables and intervention and control groups were tested using Chi-square tests *(X*^2^) (or Fisher Exact Tests where appropriate). Significance was set as *p* < 0.05.

**Table 4 ijerph-16-01305-t004:** Baseline outcome measures (*n* = 32).

Variable of Interest	Total Participants (*n* = 32)	IG *n* = 15 (46.88%)	CG *n* = 17 (53.13%)	Statistic	*p*-Value
**Usage mobile phone**					
Mobile function used most				*X*^2^ = 2.89	*p* = 0.409
Calls (*n* = 31)	26 (83.87%)	13 (92.86%)	13 (76.5%)		
Frequency network contact				*X*^2^ = 1.41	*p* = 0.703
with mobile phone					
Weekly (*n* = 31)	17 (54.84%)	7 (50.0%)	10 (58.8%)		
**Cognition (YSQ-SF domain Disconnection and Rejection)**					
Total YSQ-SF/150, α 0.75	78.38 ± 25.06	83.53 ± 19.30	73.82 ± 29.05	*U* = 0.46	*p* = 0.478
Emotional dep./30, α 0.80	16.91 ± 6.16	18.93 ± 5.65	15.12 ± 6.18	*U* = −1.52	*p* = 0.128
Social Isolation/30, α 0.78	16.72 ± 5.99	17.87 ± 5.68	15.71 ± 6.23	*U* = −0.57	*p* = 0.570
Abandonment/30, α 0.81	15.28 ± 7.28	16.80 ± 5.98	13.94 ± 8.21	*U* = −1.03	*p* = 0.305
Defectiveness/30, α 0.80	14.78 ± 5.66	14.80 ± 5.19	14.76 ± 6.21	*U* = −0.11	*p* = 0.910
Mistrust/25, α 0.78	14.69 ± 5.50	15.13 ± 5.28	14.29 ± 5.82	*U* = −0.30	*p* = 0.762
**Loneliness (DJGLS)**					
Loneliness total/6, α 0.61	3.56 ± 1.24	3.53 ± 1.30	3.59 ± 1.23	*U* = −0.35	*p* = 0.727
Emotional loneliness/3, α 0.65	1.63 ± 0.83	1.73 ± 0.80	1.53 ± 0.87	*U* = 0.48	*p* = 0.477
Social loneliness/3, α 0.55	1.94 ± 0.72	1.80 ± 0.68	2.06 ± 0.75	*U* = −1.01	*p* = 0.314
**Mental well-being (WHO-5)**					
WHO-5/25, α 0.81	16.22 ± 5.32	15.07 ± 6.87	17.24 ± 3.35	*U =* −1.01	*p =* 0.314

CG: Control group; dep: Deprivation; DJGLS: de Jong Gierveld loneliness scale; IG: Intervention group; WHO-5: WHO-five Well-being Index; YSQ-SF: Young Schema Questionnaire (Short form); Note: Differences between intervention and control groups for usage of mobile phone were tested using Chi-square tests *(X*^2^) and cognition, loneliness, and mental well-being were tested using non-parametric test (Mann-Whitney *U*; Significance was set as *p* < 0.05.

**Table 5 ijerph-16-01305-t005:** Pre- and Post-intervention comparisons of social cognitions, loneliness, and mental well-being.

	Scale Item	m, sd (T_0_)	m, sd (T_1_)	m, sd (T_2_)	*p*-Value (T_0_–T_2_) (F)	*p*-Value T_0_–T_1_ (W)	*p*-Value T_1_–T_2_ (W)	*p*-Value T_0_–T_2_ (W)
IG (T_0_ *n* = 15, T_1_ and _2_, *n* = 13)	**Social Cognition (YSQ-SF)**							
	Total YSQ-SF/150	83.53 ± 19.30	53.85 ± 25.30	52.62 ± 15.99	0.008 *	0.019 *	0.969	0.003 *
	Emotional Dep./30	18.93 ± 5.65	12.31 ± 8.29	10.23 ± 2.46	0.003 *	0.014 *	0.582	0.003 *
	Social Isolation/30	17.87 ± 5.68	10.38 ± 4.33	10.23 ± 3.59	<0.001 *	0.002 *	0.755	0.002 *
	Abandonment/30	16.80 ± 5.98	10.23 ± 5.72	10.69 ± 5.17	0.193	0.037 *	0.655	0.041 *
	Mistrust/30	15.13 ± 5.28	12.92 ± 6.65	12.08 ± 3.93	0.787	0.576	0.688	0.263
	Defectiveness/30	14.80 ± 5.19	8.00 ± 4.08	9.38 ± 3.36	0.009 *	0.007 *	0.089	0.025 *
	**Loneliness (DJGLS)**							
	Total loneliness/6	3.53 ± 1.30	2.31 ± 1.49	1.38 ± 1.33	0.001 *	0.031 *	0.028 *	0.003 *
	Social loneliness/3	1.80 ± 0.68	1.69 ± 0.75	1.08 ± 0.86	0.086	0.655	0.054	0.058
	Emotional loneliness/3	1.73 ± 0.80	0.62 ± 0.96	0.31 ± 0.63	<0.001 *	0.023 *	0.157	0.003 *
	**Mental well-being (WHO-5)**							
	WHO-5/25	15.07 ± 6.87	17.15 ± 6.31	16.54 ± 4.54	0.341	0.161	0.504	0.643
CG (T_0_ –T_2_ *n* = 17)	**Social Cognition (YSQ-SF)**							
	Total YSQ-SF/150	73.82 ± 29.05	70.59 ± 22.15	78.00 ± 14.77	0.275	0.642	0.170	0.413
	Emotional Dep./30	15.12 ± 6.18	14.00 ± 5.40	16.06 ± 4.18	0.476	0.534	0.195	0.704
	Abandonment/30	13.94 ± 8.21	13.82 ± 5.58	14.41 ± 5.08	0.570	0.887	0.670	0.932
	Social Isolation/30	15.71 ± 6.23	14.41 ± 5.01	16.47 ± 3.69	0.279	0.377	0.129	0.660
	Mistrust/30	14.29 ± 5.82	15.94 ± 5.32	17.65 ± 4.86	0.566	0.225	0.147	0.117
	Defectiveness/30	14.76 ± 6.21	12.41 ± 5.43	13.41 ± 5.35	0.729	0.116	0.529	0.378
	**Loneliness (DJGLS)**							
	Total loneliness/6	3.59 ± 1.23	2.47 ± 2.10	4.00 ± 1.32	0.064	0.086	0.011 *	0.282
	Social loneliness/3	2.06 ± 0.75	1.47 ± 1.18	1.88 ± 0.57	0.414	0.080	0.176	0.386
	Emotional loneliness/3	1.53 ± 0.87	1.00 ± 1.12	2.12 ± 0.70	0.002 *	0.075	0.003 *	0.704
	**Mental well-being (WHO-5)**							
	WHO-5/25	17.24 ± 3.35	16.76 ± 4.70	16.47 ± 4.00	0.591	0.584	0.699	0.413

CG: Control group; DJGLS: de Jong Gierveld loneliness scale; Dep.: Deprivation; F: Friedman test; IG: Intervention group; T: Time; WHO-5: WHO (five) Well-being Index; W: Wilcoxon Signed Rank test; YSQ-SF: Young Schema Questionnaire (Short form); Note: Differences between baseline/pre-intervention (T_0_), post-intervention (T_1_), post-maintenance (T_2_) were tested using non-parametric Friedman test; Differences between T_0_–T_1_, T_1_–T_2_, T_0_–T_2_ were tested using Wilcoxon mean signed test (*W*); Significance was set as *p* < 0.05 *.

**Table 6 ijerph-16-01305-t006:** Between group comparisons of changes through the stages for social cognitions, loneliness, and mental well-being.

	T_0_–T_1_			T_1_–T_2_			T_0_–T_2_			
Scale Item	IG (*n* = 13) (M, Sd of Differences)	CG (*n* = 17) (M, Sd of Differences)	*p*-Value	IG (*n* = 13) (M, Sd of Differences)	CG (*n* = 17) (M, Sd of Differences)	*p*-Value	IG (*n* = 13) (M, Sd of Differences)	CG (*n* = 17) (M, Sd of Differences)	*p*-Value	Effect Size (Hedge’s g)
**Social Cognition (YSQ-SF)**										
Total YSQ-SF	28.31 ± 34.13	3.24 ± 25.94	0.065	1.23 ± 15.79	−7.41 ± 21.56	0.133	−29.54 ± 25.18	4.18 ± 29.13	0.006 *	1.23 ***
Emotional Dep.	6.69 ± 7.58	1.12 ± 7.7,5	0.025 *	2.08 ± 8.10	−2.06 ± 6.80	0.229	−8.77 ± 6.58	0.94 ± 8.64	0.002 *	1.24 ***
Abandonment	6.69 ± 7.58	1.12 ± 7.75	0.025 *	−0.46 ± 4.60	−0.59 ± 7.29	0.805	−5.77 ± 7.54	0.47 ± 8.78	0.053	0.75 **
Social Isolation	7.08 ± 5.62	1.29 ± 6.08	0.015 *	0.15 ± 3.18	−2.06 ± 5.86	0.183	−7.23 ± 5.73	0.76 ± 6.58	0.002 *	1.28 ***
Mistrust	1.92 ± 10.16	−1.65 ± 6.73	0.300	0.85 ± 4.78	−1.71 ± 4.95	0.245	−2.77 ± 7.93	3.35 ± 7.75	0.048 *	0.78 **
Defectiveness	6.38 ± 5.85	2.35 ± 5.93	0.059	−1.38 ± 2.63	−1.00 ± 6.02	0.680	−5.00 ± 6.58	−1.35 ± 6.22	0.123	0.57 **
**Loneliness (DJGLS)**										
Total loneliness	1.15 ± 1.57	1.12 ± 2.34	0.902	0.92 ± 1.26	−1.53 ± 2.00	0.001 *	−2.08 ± 1.75	0.41 ± 1.58	<0.001 *	1.50 ***
Social loneliness	0.08 ± 0.64	0.59 ± 1.50	0.408	0.62 ± 1.04	−0.41 ± 1.18	0.028 *	−0.69 ± 1.18	−0.18 ± 1.01	0.229	0.47 *
Emotional loneliness	1.08 ± 1.26	0.53 ± 1.18	0.157	0.31 ± 0.75	−1.12 ± 1.11	0.001 *	−1.38 ± 0.87	0.59 ± 0.87	<0.001 *	2.26 ***
**Mental well-being (WHO-5)**										
WHO-5	−1.92 ± 4.21	0.47 ± 4.26	0.113	0.62 ± 4.94	0.29 ± 4.43	0.621	1.31 ± 5.91	−0.76 ± 4.40	0.363	0.41 *

CG: Control group; Dep.: Deprivation; DJGLS: de Jong Gierveld loneliness scale; IG: Intervention group; WHO-5: WHO (five) Well-being Index; YSQ-SF: Young Schema Questionnaire (Short form); Note: Mean and standard deviations of differences between pre- and post-intervention (T_0_–T_1_) and post-intervention and post-maintenance (T_1_–T_2_) between IG and CG were tested using non-parametric test (Mann-Whitney-U test); Significance was set as *p* < 0.05 *. Effect sizes calculated for T_0_–T_2_ using Hedges’ g.* 0.2: small effect size, ** 0.5 medium effect size, *** 0.8 large effect size.

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
