# Peer review of "An Evaluation of a Low-Intensity Cognitive Behavioral Therapy mHealth-Supported Intervention to Reduce Loneliness in Older People"

_ijerph, 2019, doi:10.3390/ijerph16071305_

Reviewer 1 Report

This is a well written and set out paper, that gave me no difficulties as a reader or understanding the results.

I commend the study design where we seldom see formal time spent in supporting individuals to gain confidence in the use of a technology solution.

My only point to give greater clarity and to set expectations would to change the title of the article By adding the word "mHealth SUPPORTED intervention" as this better describes the intervention clearly for readers.... 

Author Response

This is a well written and set out paper, that gave me no difficulties as a reader or understanding the results.I commend the study design where we seldom see formal time spent in supporting individuals to gain confidence in the use of a technology solution.

Thank you

My only point to give greater clarity and to set expectations would to change the title of the article By adding the word "mHealth SUPPORTED intervention" as this better describes the intervention clearly for readers.... 

Title adjusted to include “supported” and added throughout the text. 

Reviewer 2 Report

Valuable paper about the use of Whatsapp as a tool to face isolation.

Being worth to be published, please address some issues:

1) Introduce in the abstract some quantitative results that help the reader to understand the magnitude of your work

2) Please, make more explicit the relationship between health and isolation, as this is one of the top determinants for health outcomes. Please include a specific reference about this.

3) Please include in the paper references to other studies using Whatsapp to face isolation. It is also recommended to make a comparison between your intervention and others making use of whatsapp or any other social media

3) In the discussion, justify whether your results can be scaled to other settings or regions in the world. How that should be customised to keep or improve your results?

4) High percentage of eligible people did not participate in the study. Reasons are provided in the text but authors also encouraged to raise how they consider to improve engagement rate to these intervention programs and also consider if the ones rejecting to participate were the most critical ones from the health outcomes perspective.

Author Response

1) Introduce in the abstract some quantitative results that help the reader to understand the magnitude of your work

Introduced into the abstract the values of three key markers. The following sentence was expanded as below:

There were significant changes in social cognition (YSQ-SF T0–T1–T2, X2=9.69, p=.008) and loneliness levels (total loneliness T0–T1–T2, X2=14.62, p=.001), and an increase in WhatsApp usage (T0 = 26% vs. T1 = 60%, X2=15.22, p=.019).

2) Please, make more explicit the relationship between health and isolation, as this is one of the top determinants for health outcomes. Please include a specific reference about this.

A sentence has been included at the end of the first paragraph of the introduction and background showing the influence of loneliness on mortality:

Loneliness has pathophysiological effects on older persons which increase mortality 11.

3) Please include in the paper references to other studies using Whatsapp to face isolation. It is also recommended to make a comparison between your intervention and others making use of whatsapp or any other social media

The study is unique in its offering of LI-CBT supported by WhatsApp. Numerous education linked intervention studies have demonstrated the effectiveness of WhatsApp, but there is a paucity of intervention studies using social media as a platform to address loneliness in older persons.

The following sentence has been added to the 2nd paragraph of the introduction and background:

There are a plethora of primary studies, and systematic reviews addressing loneliness in older persons 5,12,13.

Further in this paragraph the following was added:

Despite studies examining older persons’ existing engagement with social media and its effects on loneliness 21,31 a gap exists for intervention studies that specifically target loneliness using social media. Chipps et al. 14 in their umbrella review of systematic reviews examining eInterventions targeting loneliness in older persons failed to identify any mHealth interventions in the included primary studies. Chen and Schulz 12 in their systematic review of communication technology interventions, suggested further studies in the fields of loneliness and social media to identify the opportunities to reduce loneliness in older persons.

3) In the discussion, justify whether your results can be scaled to other settings or regions in the world. How that should be customised to keep or improve your results?

A phrase has been added to the discussion to qualify that it can be universally applied, but that data costs might need less consideration in resourced settings.  Customisation is also addressed as shown below:

 …and can be used in other resource restricted settings and non-resource restricted settings where consideration to data costs might be less significant and allow for the use of all WhatsApp functions. Customization of the intervention is suggested in tailoring the positively worded MIM to be culturally and contextually specific. 

4) High percentage of eligible people did not participate in the study. Reasons are provided in the text but authors also encouraged to raise how they consider to improve engagement rate to these intervention programs and also consider if the ones rejecting to participate were the most critical ones from the health outcomes perspective.

In the discussion failure to enrol is discussed in the following line:

In future at the invitation to the intervention a period of experimentation with a smartphone in a low risk setting has the possibility of increasing enrolment.

The question about the critical markers of the eligible participants’ failing to enrol shows they had similar mean scores to the participating persons.

Reviewer 3 Report

The paper presents an intervetion on using mHealth (specifically WhatsApp) to reduce loneliness in older persons. The paper is well write and is easy to read. The desing of the experiment is appropriate and the results are interesnting.

The are, however, a couple of aspects that could be improved.

First, the papers lack a related works sections. There are a lot of works that address loneliness in the elder, even in the mHealth area. How is this work different? How are their results compared with the others?

And second, I think it could be interesting to have additional information about how the elders used WhatsApp. The paper states that some of the elders already used a mobile phone frecuently, mainly to call family an friends. Did some of the elders used WhatsApp before the experiment? It has some effect on the results? Also, during the experiment how was WhatsApp used? To write messages, to make video calls, to send audio notes? Whio where the elders communicating with? Previously existing contacts or new contacr. In general more information about the phone use during the experiment would be interesting.

Author Response

First, the papers lack a related works sections. There are a lot of works that address loneliness in the elder, even in the mHealth area. How is this work different? How are their results compared with the others?

The study is unique in its offering of LI-CBT supported by WhatsApp. Numerous education linked intervention studies have demonstrated the effectiveness of WhatsApp, but there is a paucity of intervention studies using social media as a platform to address loneliness in older persons. Chipps et al., (2017) in their umbrella review of systematic reviews examining eInterventions targeting loneliness in older persons failed to identify any mHealth interventions in the included primary studies.

Reference has been made to Chen and Schulz (2016) suggestion that the gap is addressed.

Additional information reflects this in text:

Despite studies examining older persons’ existing engagement with social media and its effects on loneliness 21,31 a gap exists for intervention studies that specifically target loneliness using social media. Chipps et al. 14 in their umbrella review of systematic reviews examining eInterventions targeting loneliness in older persons failed to identify any mHealth interventions in the included primary studies. Chen and Schulz 12 in their systematic review of communication technology interventions, suggested further studies in the fields of loneliness and social media to identify the opportunities to reduce loneliness in older persons

And second, I think it could be interesting to have additional information about how the elders used WhatsApp. The paper states that some of the elders already used a mobile phone frecuently, mainly to call family an friends. Did some of the elders used WhatsApp before the experiment? It has some effect on the results? Also, during the experiment how was WhatsApp used? To write messages, to make video calls, to send audio notes? Whio where the elders communicating with? Previously existing contacts or new contacr. In general more information about the phone use during the experiment would be interesting.

At commencement the majority of participants owned feature (basic) mobile phones which do not support internet linked applications. Participants who owned smartphones continued to use them as feature phones. The researchers only had access to the mLINCC WhatsApp group which showed participants active engagement with all functions on WhatsApp, but had more difficulty to create voice messages.

Additions have been added to the description of the intervention and the results sections.

mLINCC

….All participants in the intervention group (IG) were added to mLINCC WhatsApp group, which was a closed group and at cross-over a separate WhatsApp group (mLINCC 2) for the control group (CG). Apart from the closed groups, participants were able to utilize the smartphone and communication on mLINCC as they chose and there was no incentive for increased use. The data package selected to offer the greatest financial sustainability post intervention, allowed for any form of WhatsApp interaction apart from WhatsApp calls. No participants had previously used WhatsApp; in the IG apart from one participant all owned feature (basic) mobile phones which did not support such applications as WhatsApp, predominantly making calls and the participant who owned a smartphone used it as a feature (basic) mobile phone

Results

Monitoring of participants’ smartphone activity was only possible through the WhatsApp group (mLINCC), but not through their encrypted messages. On mLINCC there was evidence of participants’ active use of all functions offered by WhatsApp except WhatsApp voice calls due to the data package. It appeared to take more practice to create voice messages due to the co-ordination required in the process.

Reviewer 4 Report

The authors report on a study that uses WhatsApp as an intervention to reduce loneliness in older persons. The paper is interesting and presents the study adequately. However, there are some concerns with the research design, and the discussion and the conclusion sections need some upgrade.

Table 1 is difficult to interpret; how should the third column be read, and how are the rows of the left side of the table aligned with the content of the column on the right side?

In Table 3, the authors present the ethnic group of subjects in the demographics-section. The usefulness of this variable is questionable, and this variable is not further used in the study. Further, there is no mechanism that would suggest a direct impact of ethnicity on loneliness or other outcomes. 

As a further note, dividing the number of participants into sub-classes, as outlined in Table 3, can result in problems for statistical analysis, as the number of participants is rather low. Further, is the p-value of 1.0 for "children" real? Please check. Given the rather low number of participants, the number of tests (outcome measures) is high, which could give unwanted effects for the analysis.

The authors mention that only IG uses WhatsApp during the study. How do the authors avoid a possible bias that users of WhatsApp perform extra well (a kind of placebo-effect)? I understand that CG receives WhatsApp only after the study. As an alternative, the CG could use the phones to play an "irrelevant" game on the phone that does not have an impact on the outcomes. 

The authors do not describe which functionality of WhatsApp is used in the intervention. The functionality used in the intervention should be presented to some detail, as the functionality has a major impact on parameters such as loneliness (as discussed in Section 5). One can design the use of WhatsApp in a way that the app is used as a communication channel in a constant sized group or one could give incentives to increase the size of the group of people the subjects communicate with. This is, however, not described in the paper. Further, such issues should be discussed in Section 5. Possibly, the impact of using a different set of functionality could be higher than the constructs of mistrust or defectiveness as discussed by the authors. Therefore, the set of functionality would be more than just a limitation, as suggested by the authors in the limitations-section under "thirdly", on "page2,line64" in the manuscript.

The conclusion section is quite generic. As a recommendation, the authors should add some words about the use of interventions such as WhatsApp in the future, and point out which further research is suggested considering the results of the study.

Author Response

Table 1 is difficult to interpret; how should the third column be read, and how are the rows of the left side of the table aligned with the content of the column on the right side?

Table 1 has been re-structured. 

In Table 3, the authors present the ethnic group of subjects in the demographics-section. The usefulness of this variable is questionable, and this variable is not further used in the study. Further, there is no mechanism that would suggest a direct impact of ethnicity on loneliness or other outcomes.  

Ethnicity was included due to South Africa’s history of racial division and its reflection in various studies in South Africa. Even though not directly discussed in this paper the article referenced in the discussion section discussing the glue that binds society has reference to racial differences. 

As a further note, dividing the number of participants into sub-classes, as outlined in Table 3, can result in problems for statistical analysis, as the number of participants is rather low. Further, is the p-value of 1.0 for "children" real? Please check. Given the rather low number of participants, the number of tests (outcome measures) is high, which could give unwanted effects for the analysis. 

As a limitation to the study the authors recognise that the sample is small. However due to the randomisation of participants into control and intervention groups the analysis needs to occur with two separate groups of participants.

p-value checked and changed to.589 (one sided). Thank you. 

The authors mention that only IG uses WhatsApp during the study. How do the authors avoid a possible bias that users of WhatsApp perform extra well (a kind of placebo-effect)? I understand that CG receives WhatsApp only after the study. As an alternative, the CG could use the phones to play an "irrelevant" game on the phone that does not have an impact on the outcomes.  

The CG did not receive their smartphones until the crossover point, hence had no opportunity to experiment with their phones. A sentence has been added to the methodology to clarify this.

… receiving their smartphones and all resources at this cross-over point.

The authors do not describe which functionality of WhatsApp is used in the intervention. The functionality used in the intervention should be presented to some detail, as the functionality has a major impact on parameters such as loneliness (as discussed in Section 5). One can design the use of WhatsApp in a way that the app is used as a communication channel in a constant sized group or one could give incentives to increase the size of the group of people the subjects communicate with. This is, however, not described in the paper. Further, such issues should be discussed in Section 5. Possibly, the impact of using a different set of functionality could be higher than the constructs of mistrust or defectiveness as discussed by the authors. Therefore, the set of functionality would be more than just a limitation, as suggested by the authors in the limitations-section under "thirdly", on "page2,line64" in the manuscript. 

The functionality has been added into section 2 in the description of the intervention

All participants in the intervention group (IG) were added to mLINCC WhatsApp group, which was a closed group and later a separate WhatsApp group (mLINCC 2) for the control group (CG). Participants were able to utilize the smartphone and communication on mLINCC as they chose and there was no incentive for increased use. The data package selected to offer the greatest financial sustainability post intervention in a resource restricted setting, allowed for any form of WhatsApp interaction apart from WhatsApp calls.  

The conclusion section is quite generic. As a recommendation, the authors should add some words about the use of interventions such as WhatsApp in the future, and point out which further research is suggested considering the results of the study. 

The conclusion has been adjusted and added a section outlining recommendations. They read as follows:

Conclusion

The unique LI-CBT mHealth supported by the social networking platform of WhatsApp (mLINCC) incorporated face-to-face development of technological readiness in smartphone usage, psycho-education, and the delivery of positively worded mobile instant messages. Overall the intervention showed significant improvements in maladaptive cognitions, and loneliness, highlighting the role of cognitive appraisals in the subjective experience of loneliness. In addition, the unique use of WhatsApp and the crafting of a LI-CBT mHealth supported intervention that addressed technological readiness in smartphone usage mitigated against loneliness experienced by cognitively intact older persons living in residential care settings. The study demonstrated both the possibility of the delivery of LI-CBT by a non-psychologist in a low resourced setting in Africa and demonstrated the ability of older persons to learn smartphone technology facilitating e-inclusion.

7.       Recommendations

Residential care facilities implement measures to facilitate positive cognitive appraisals of relationships and contexts. In order to provide for greater sustainability such activities should be built into daily life activities rather than predetermined programs.

A repeat study with a larger sample for a longer period or involving community dwelling older persons.

Further studies on the development of trust amongst older persons as a measure to increase enrolment in intervention programs and decrease social loneliness.

Round  2

Reviewer 4 Report

The authors have made an attempt to address the comments. However, I do not understand the meaning of some of the newly introduced sentences. Please revise and rephrase these sentences. E.g., page 7: what is the "data package"? It seems that there is some context that is not explained to the readers (possibly, the business model of the local telecom provider?). The sentence "It appeared to take more practice ..." is incomprehensible. 

New sentences in Section 5 are incomprehensible. As a note, data costs might no longer be an issue, as business models for mobile data change rapidly.

Also sentences in Sections 6 and 7 need to be rephrased. Several of these sentences are incomprehensible and incomplete: e.g., sentence "A repeat study ...". The new recommendation section should be more specific and also include more details.

Regarding Table 3: I recommend to split the demographics and the outcome measures into two separate tables.

Regarding the ethnic group variable: if the ethnic group implies social differences, as you comment, parameters indicating these social indicators should be outlined instead (e.g., income, wealth, type of dwelling before going to res.).

Given the relatively low number of subjects in the study, the number of demographics variables  and outcome measures is rather high; this can have an impact on significance.

Author Response

Please revise and rephrase these sentences. E.g., page 7: what is the "data package"? It seems that there is some context that is not explained to the readers (possibly, the business model of the local telecom provider?). The sentence "It appeared to take more practice ..." is incomprehensible.

Sentences have been re-phrased to improve clarity.

… On mLINCC there was evidence of participants’ active use of all functions offered by the select WhatsApp data package (excluded WhatsApp calls), except for the function of creating a voice message. While recording a voice message and the simultaneous requirement of holding the voice message icon appeared to present co-ordination challenges.

New sentences in Section 5 are incomprehensible. As a note, data costs might no longer be an issue, as business models for mobile data change rapidly.

Noted and deleted

Also sentences in Sections 6 and 7 need to be rephrased. Several of these sentences are incomprehensible and incomplete: e.g., sentence "A repeat study ...". The new recommendation section should be more specific and also include more details.

Sentences have been rephrased.

It is recommended that residential care facilities implement measures to facilitate residents’ positive cognitive appraisals of relationships and contexts such as regular psychoeducation sessions. It is further recommended that the study is repeated with a larger sample, over for a longer period, possibly six months and involves community dwelling older persons. In particular this should be conducted in Africa, where there is a paucity of comparable loneliness studies. In addition, further studies on the development of trust amongst older persons could result in a measure to increase enrolment in intervention programs targeting loneliness

Regarding Table 3: I recommend to split the demographics and the outcome measures into two separate tables. 

Thank you for the suggestion. The table has been split and in text references, refer to table 3 or 4 respectively. 

Regarding the ethnic group variable: if the ethnic group implies social differences, as you comment, parameters indicating these social indicators should be outlined instead (e.g., income, wealth, type of dwelling before going to res.). 

Thank you for your comment. Race has been deleted from Table 3. 

Given the relatively low number of subjects in the study, the number of demographics variables  and outcome measures is rather high; this can have an impact on significance. 

Comment noted and added to limitations